# Inner Monologue: Embodied Reasoning through Planning with Language Models

**Wenlong Huang**[†]**, Fei Xia**[†]**, Ted Xiao**[†]**, Harris Chan, Jacky Liang, Pete Florence,**
**Andy Zeng, Jonathan Tompson, Igor Mordatch, Yevgen Chebotar, Pierre Sermanet,**
**Noah Brown, Tomas Jackson, Linda Luu, Sergey Levine, Karol Hausman, Brian Ichter**

Robotics at Google, [†] equal contribution and alphabetically listed
Project website: https://innermonologue.github.io

**Abstract:** Recent works have shown how the reasoning capabilities of Large Language Models (LLMs) can be applied to domains beyond natural language processing, such as planning and interaction for robots. These embodied problems require an agent to understand many semantic aspects of the world: the repertoire of skills available, how these skills influence the world, and how changes to the world map back to the language. LLMs planning in embodied environments need to consider not just what skills to do, but also how and when to do them - answers that change over time in response to the agent's own choices. In this work, we investigate to what extent LLMs used in such embodied contexts can reason over sources of feedback provided through natural language, without any additional training. We propose that by leveraging environment feedback, LLMs are able to form an *inner monologue* that allows them to more richly process and plan in robotic control scenarios. We investigate a variety of sources of feedback, such as success detection, scene description, and human interaction. We find that closed-loop language feedback significantly improves high-level instruction completion on three domains, including simulated and real table top rearrangement tasks and long-horizon mobile manipulation tasks in a kitchen environment in the real world.

## 1   Introduction

Intelligent and flexible embodied interaction requires robots to be able to deploy large repertoires of basic behaviors in appropriate ways, sequence these behaviors as needed for long horizon tasks, and also recognize when to switch to a different approach if a particular behavior or plan is unsuccessful. High-level planning, perceptual feedback, and low-level control are just a few of the sub-tasks that would need to be seamlessly combined together to perform the sort of reasoning required for an embodied agent, such as a robot, to intelligently act in the world. While conventionally these challenges have been approached from the perspective of planning (e.g., TAMP [1]) or hierarchical learning (e.g., HRL [2]), effective high-level reasoning about complex tasks also requires semantic knowledge and understanding of the world.

One of the remarkable observations in recent machine learning research is that large language models (LLMs) can not only generate fluent textual descriptions, but also appear to have rich internalized knowledge about the world [3, 4, 5, 6, 7]. When appropriately conditioned (e.g., prompted), they can even carry out some degree of deduction and respond to questions that appear to require reasoning and inference [8, 9, 10, 11, 12, 13]. This raises an intriguing possibility: beyond their ability to interpret natural language instructions, can language models further serve as reasoning models that combine multiple sources of feedback and become interactive problem solvers for embodied tasks, such as robotic manipulation?

Prior studies show that language helps humans internalize our knowledge and perform complex relational reasoning through *thinking in language* [14, 15, 16, 17, 18]. Imagine the "inner monologue" that happens when a person tries to solve some task: "I have to unlock the door; let me try to pick up the key and put it in the lock... no, wait, it doesn't fit, I'll try another one... that one worked, now I can turn the key." The thought process in this case involves choices about the best immediate action to solve the high-level task ("pick up the key"), observations about the outcomes of attempted actions ("it doesn't fit"), and corrective actions that are taken in response to these observations ("I'll try another one"). Inspired by the human thought process, we propose that such an inner monologue is a natural framework for incorporating feedback for LLMs.

6th Conference on Robot Learning (CoRL 2022), Auckland, New Zealand.

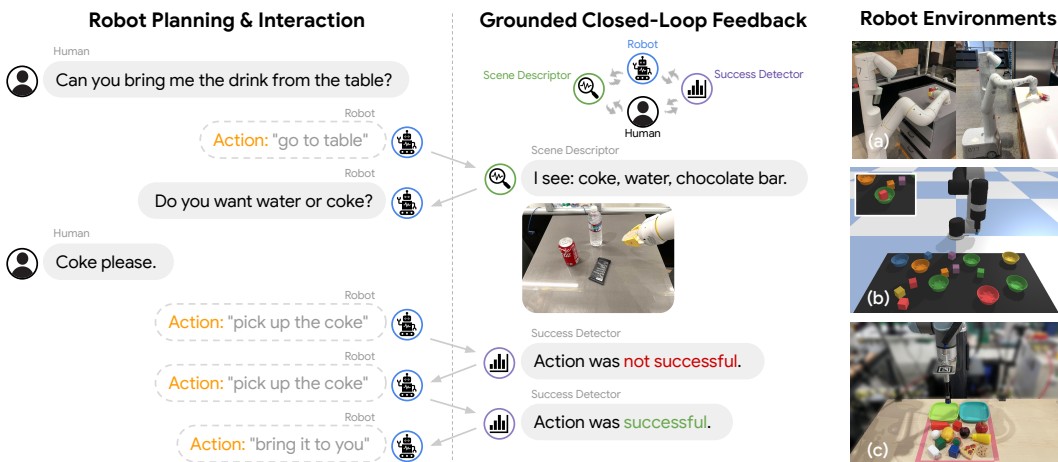

**Figure 1: Inner Monologue** enables grounded closed-loop feedback for robot planning with large language models by leveraging a collection of perception models (e.g., scene descriptors and success detectors) in tandem with pretrained language-conditioned robot skills. Experiments show our system can reason and replan to accomplish complex long-horizon tasks for (a) mobile manipulation and (b,c) tabletop manipulation in both simulated and real settings.

Our work studies these questions by combining LLMs with various sources of textual feedback, only utilizing few-shot prompting without any additional training. We observe that similarly to recent work [19], natural language provides a universal and interpretable interface for such grounding of model communication and allows them to incorporate their conclusions in an overarching inner monologue driven by a language model. While prior work has investigated using language models as planners [20, 21] or incorporating multimodal-informed perception through language [19], to the best of our knowledge no work has studied the critical link of not only planning with language, but also informing *embodied feedback with language*, which we investigate in this work.

Specifically, we study methods and sources of feedback for closing the agent-environment loop via an inner monologue and their impact on downstream execution success and new capabilities arising from such interaction. In particular, we combine multiple perception models that perform various tasks such as language-conditioned semantic classification or language-based scene description, together with feedback provided by a human user that the robot is cooperating with. To execute the commands given by a user, the actions are chosen from a set of pre-trained robotic manipulation skills together with their textual descriptions that can be invoked by a language model. Our proposed system Inner Monologue chains together these various components (perception models, robotic skills, and human feedback) in a shared language prompt, enabling it to successfully perform user instructions.

Finally, we show that Inner Monologue, without requiring additional training beyond a frozen language model and pre-trained robotic skills, can accomplish complex, long-horizon, and unseen tasks in simulation as well as on two real-world robotic platforms. Notably, we show that it can efficiently retry under observed stochastic failure, replan under systematic infeasibility, or request human feedback for ambiguous queries, resulting in significantly improved performance in dynamical environments. As a demonstration of the versatility of LLMs and grounded closed-loop feedback, we additionally show several surprising capabilities emerging from the inner monologue formulation, including continued adaptation to new instructions, self-proposed goals, interactive scene understanding, multilingual interactions, and more.

## 2   Related Work

**Task and Motion Planning.**   Task and motion planning [22, 23] requires simultaneously solving a high-level, discrete task planning problem [24, 25, 26], and a low-level, continuous motion planning problem [27]. Traditionally, this problem has been solved through optimization [28, 29] or symbolic reasoning [24, 26], but more recently machine learning has been applied to aspects of the problem via learned representations, learned task-primitives, and more [30, 31, 32, 33, 34, 35, 36, 37, 38]. Some works utilize language for planning and grounding [39, 40, 41, 42, 43, 44]. Others have approached the problem through hierarchical learning [45, 46, 34, 47, 48, 49, 50]. In this work, we leverage pre-trained LLMs and their semantic knowledge, along with trained low-level skills, to find feasible plans.

**Task Planning with Language Models.**   Various prior works have explored using language as a space for planning [51, 52, 53, 20, 54, 21]. Some methods use prompt structure, self-talk, or discussion between

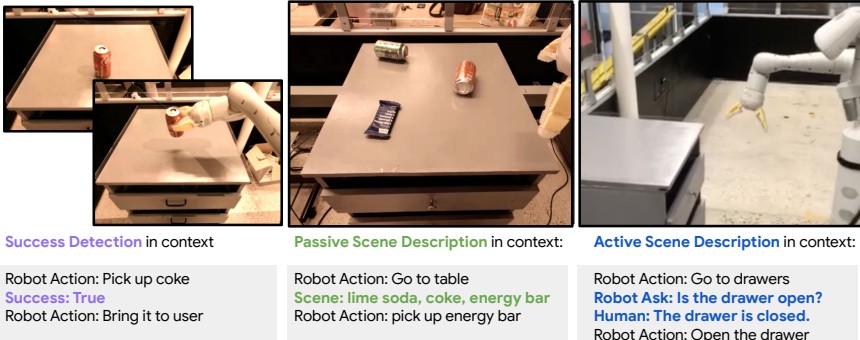

**Figure 2:** Various types of textual feedback. **Success Detection** gives task-specific task completion information, **Passive Scene Description** gives structured semantic scene information at every planning step, and **Active Scene Description** gives unstructured semantic information only when queried by the LLM planner.

experts to reason about plans or semantic concepts [55, 19, 10, 11]. Similar to ours are recent task planning approaches that leverage pre-trained autoregressive LLMs to decompose abstract, high-level instructions into a sequence of low-level steps executable by an agent [20, 21] in a zero-shot manner. Specifically, Huang et al. [20] prompt GPT-3 [9] and Codex [56] to generate action plans for embodied agents, where each action step is semantically translated to an admissible action with a Sentence-RoBERTa model [57, 58]. SayCan [21] instead grounds the actions by multiplying each candidate action's probability under FLAN [59] with the action's value function, which serves as a proxy for affordance [34]. However, both approaches effectively produce the plan while assuming that each proposed step is executed successfully by the agent. As a result, these approaches may not be robust in handling intermediate failures in dynamic environments or with poor lower level policies. We explore in Inner Monologue ways to incorporate grounded feedback from the environment into the LLM as we produce each step in the plan.

**Fusing Vision, Language, and Control in Robotics.** Various works have investigated strategies for the challenging problem of fusing vision, language, and control [60, 61, 62, 63, 64, 65, 66]. Some works have been trained directly for language-based interaction in robotic tasks [67, 68, 69, 70]. Recent large visual-language models (e.g., CLIP [71]) have been trained on joint image(s) and corresponding text captions via variants of a masked language modeling objective [72, 73, 74, 75], a contrastive loss [76, 77, 71] or other supervised objectives[78, 79]. CLIP has been employed in several robotics and embodied settings in zero-shot manner [80], or combined with Transporter networks [81] as in CLIPort [82]. Finally, Socratic Models [19] proposes the combination of different foundation models (e.g., GPT-3 [9], ViLD [83]) and language-conditioned policies, using language as the common interface. While Socratic Models has been demonstrated on a tabletop object manipulation task, Inner Monologue examines additional challenges for robots operating in dynamic environments, which require closed-loop feedback to the planner.

## 3 Leveraging Embodied Language Feedback with Inner Monologue

We consider the setting where an embodied robotic agent attempts to perform a high-level natural language instruction $i$. This robotic agent is only capable of executing short-horizon skills from a library of previously trained policies $\pi_k \in \Pi$ with short language descriptions $\ell_k$, which may be trained with reinforcement learning or behavioral cloning. The "planner," which is a pretrained LLM [20, 21], attempts to find a sequence of skills to accomplish the instruction. To observe the environment, the planner has access to textual feedback $o$ from the environment that can be appended to the instruction or requested by the planner. Our work studies to what extent the LLM planner is able to reason over and utilize such feedback to "close the loop" with the environment and improve planning.

### 3.1 Inner Monologue

We formulate an "inner monologue" by continually injecting information from the various sources of feedback into the LLM planning language prompts as the robot interacts with the environment. While LLMs have demonstrated exceptional planning capabilities for embodied control tasks [20], prior works have found it crucial to ground LLM predictions with external components such as affordance functions [21] in order to produce useful plans that are executable by robots. However, LLMs used in this context have thus far remained one-directional – providing a list of skills, without making corrections or leveraging opportunities to replan accordingly. In contrast, Inner Monologue studies settings where grounded environment feedback

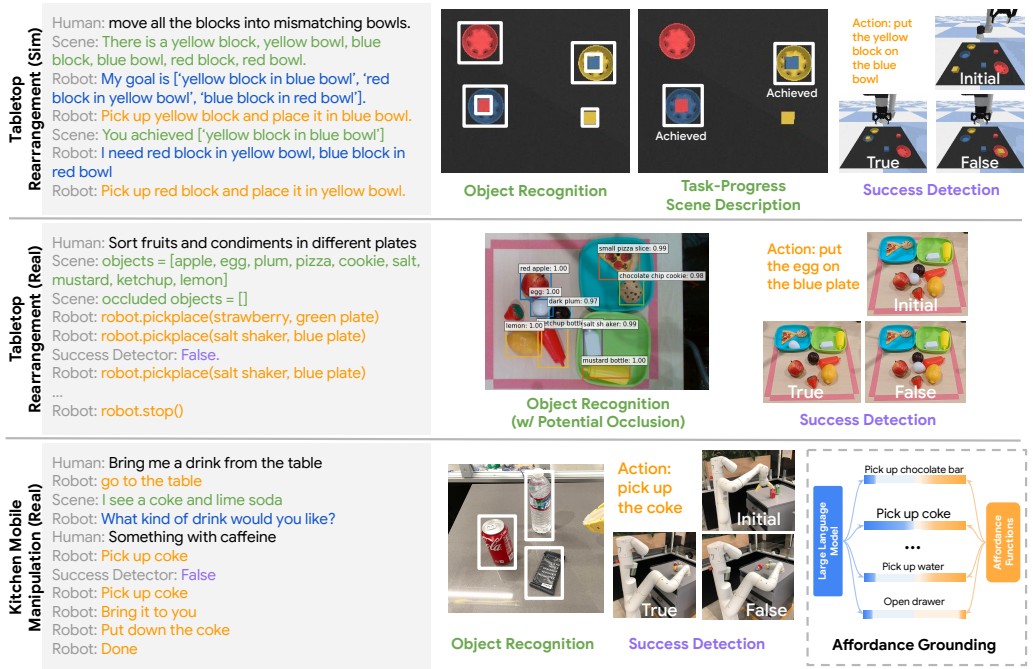

**Figure 3:** Different instantiations of Inner Monologue in three distinct domains – simulated tabletop rearrangement (*top*), real-world tabletop rearrangement (*middle*), and real-world kitchen mobile manipulation (*bottom*). Each domain uses different prompts and different feedback models. Sharing across the domains is the same Inner Monologue formulation that uses a pre-trained langauge model to take in a **human instruction** and decompose it into a sequence of **actionable steps** by the agent, while accounting for injected embodied feedback from different models, such as **object recognizers** and **success detectors**. In real-world kitchen mobile manipulation domain (*bottom*), we additionally ground the actions using pre-trained affordance functions built in [21], which do not communicate back to the language model.

is provided directly to the LLM in a closed-loop fashion. This promotes improved LLM reasoning in complex long-horizon settings, even before any external affordance-based grounding methods are applied.

Our analysis assumes textual feedback is provided to the planner, but does not assume a single specific method of fusing LLM planning with low-level robotic control or a specific method of extracting environment feedback into language. Rather than focusing on a particular algorithmic implementation, our aim is to provide a case study on the value of incorporating different types of feedback into closed-loop LLM-based planning. Thus, Inner Monologue in Sec 4 utilizes language feedback within separate systems that incorporate different LLMs, different methods of fusing planning with control, different environments and tasks, and different methods of acquiring control policies. We note that in our specific implementations of Inner Monologue, we use pre-trained LLMs for planning that are not finetuned, but rather evaluated solely with few-shot prompting; the full prompts can be found in the Appendix.

### 3.2 Sources of Feedback

In theory any type of environment feedback can inform the LLM planner, as long as it can be expressed through language. We focus on the specific forms of feedback shown in Fig 2: (1) task-specific feedback, such as success detection, and (2) scene-specific feedback (either "passive" or "active"), which describes the scene. Specific instantiations and implementation details of each type of feedback can be found in Sec 4.1, Sec 4.2, and Sec 4.3 respectively for each domain.

**Success Detection.** The *Success* feedback gives binary "yes" or "no" response in language form, specifying whether the low-level skill $\pi_k$ has succeeded. Engineered success detectors can operate on ground-truth state in simulation, while learned success detectors can be trained on real examples of successes and failures in the real world [84, 85, 86, 87, 88].

**Passive Scene Description.** We refer broadly to any sources of scene feedback that are consistently and automatically injected into the LLM prompt as Passive Scene Description, which also typically follow some structure. One common type of such feedback is object recognition [89, 90, 91, 92] that returns a list of

present objects, to which we refer as *Object* feedback. We also demonstrate the use of a task-progress scene description in the simulated tabletop rearrangement environment, to which we refer as *Scene* feedback.

**Active Scene Description.** As the proactive counterpart, Active Scene Description encompasses sources of feedback that are provided directly in response to active queries by the LLM planner, which are answered either by a person, or by another pretrained model, such as a Visual Question Answering (VQA) model [93, 94, 95, 96]. Unlike the passive counterpart which are strictly structured and narrow in their scope, this feedback allows the planner to actively gather information relevant to the scene, the task, or even preferences of the user. The combined output we send to the LLM planner includes both the LLM-generated question along with the response. As we aim to investigate *whether* and *how* a LLM planner can incorporate such feedback and wish to study both structured VQA-style human feedback as well as unstructured human preferences feedback, we only consider human-provided response in this work, which we refer to as *Human* feedback.

# 4 Experimental Results

In order to study how different sources of environment feedback can support a rich inner monologue that enables complex robotic control, we study different Inner Monologue implementations in three environments, each with different LLM and different sources of feedback from the environment: 1) simulated tabletop manipulation (Sec 4.1), 2) real-world tabletop manipulation (Sec 4.2), and 3) real-world mobile manipulation in an office kitchen (Sec 4.3). For more details about the experiment setup and results, please refer to the Appendix.

## 4.1 Simulated Tabletop Rearrangement

We experiment with Ravens-based [81] environment, where a robotic arm with a gripper is tasked with rearranging blocks and bowls in some desired configuration, specified by natural language. We evaluate each method on four seen tasks and four unseen tasks, where seen tasks may be used for training (in the case of supervised baseline) or used as few-shot prompting.

This instantiation of Inner Monologue uses (i) InstructGPT [9, 97] for planning [20, 21], (ii) scripted modules to provide language feedback in the form of object recognition (*Object*), success detection (*Success*), and task-progress scene description (*Scene*), and (iii) a pre-trained language-conditioned pick-and-place primitive (similar to CLIPort [82] and Transporter Nets [81]). *Object* feedback informs the list of present objects and *Success* feedback informs the success/failure of the most recent action. However, consider the task of stacking multiple blocks, because the unfinished tower of blocks may be knocked over by the robot, it is also critical to reason about overall task progress. Therefore, task-progress scene description (*Scene*) describes the semantic sub-goals inferred by the LLM towards completing the high-level instruction that are achieved by the agent so far.

We additionally compare to a multi-task CLIPort directly trained on long-horizon task instructions. Because CLIPort is a single-step policy and does not terminate spontaneously during policy rollout, we report CLIPort evaluations with oracle termination (i.e., repeat until oracle indicates task completion) and fixed-step termination (i.e., repeat for 15 steps). To simulate real-world disturbances and evaluate the system's robustness to disturbances, we add Gaussian noise to multiple levels of the system at test time: $\mathcal{N}(0,3)$ for pixel observation, $\mathcal{N}(0,2.5)$ for policy primitive (i.e., pick-place pixel heatmaps), $\mathcal{N}(0,0.02m)$ for place locations.

| | | | | +LLM | +Inner Monologue | |
| | Tasks | CLIPort | +oracle | *Object* | *Object + Success* | *Object + Scene* |
|---|---|---|---|---|---|---|
| | "Pick and place" | 24.0% | 74.0% | 80.0% | 90.0% | **94.0%** |
| | "Stack all the blocks" | 2.0% | 32.0% | 4.0% | 10.0% | **26.0%** |
| **Seen Tasks** | "Put all the blocks on the [x] corner/side" | 2.0% | 32.0% | **30.0%** | 28.0% | **30.0%** |
| | "Put all the blocks in the [x] bowl" | 32.0% | 94.0% | 52.0% | 46.0% | **56.0%** |
| | "Put all the blocks in different corners" | 0.0% | 0.0% | 20.0% | 20.0% | **26.0%** |
| | "Put the blocks in their matching bowls" | 0.0% | 0.0% | 56.0% | 70.0% | **82.0%** |
| **Unseen Tasks** | "Put the blocks on mismatched bowls" | 0.0% | 0.0% | 62.0% | 76.0% | **86.0%** |
| | "Stack all the blocks on the [x] corner/side" | 0.0% | 0.0% | 0.0% | 4.0% | **6.0%** |

**Table 1:** Success rates averaged across 50 episodes in simulated pick-and-place. CLIPort + oracle indicates that CLIPort was provided a "termination" oracle. LLM-informed feedback effectively enable retrying/replanning in the presence of test-time disturbances, while enjoying the generalization benefits of LLMs to unseen tasks.

**Analysis.** As shown in Table 1, Inner Monologue effectively enables retrying and replanning in the face of test-time disturbances, where *Object + Scene* performs the best because of its ability to keep track of sub-goal conditions. Furthermore, this performance directly translates to unseen tasks by leveraging rich semantic knowledge of LLM. Finally, we observe that non-hierarchical and solitary systems such as CLIPort (i) struggle at generalizing to unseen long-horizon tasks under test-time disturbances, and (ii) on training tasks, an oracle is also often required to indicate task completion for good performance.

## 4.2 Real-World Tabletop Rearrangement

We evaluate Inner Monologue on a real-world robot platform designed to resemble the simulation experiments. This instantiation uses (i) InstructGPT [9, 97] for planning, (ii) MDETR [98] for open-vocab object recognition (*Object*) (iii) heuristics on the object bounding box predictions from MDETR for Success Detection (*Success*), and (iv) a suction-based pick-and-place motion primitive that uses an LLM to parse target objects from a language command (e.g., given by the planner).

We investigate two tasks: (i) a 3-block stacking task where 2 blocks are already pre-stacked, and (ii) a long-horizon sorting task to place food in one plate and condiments in another (where categorizing food versus condiments is autonomously done by the LLM planner). In additional to additional challenges of real-world perception and clutter, we artificially inject Gaussian noise into the policy actions (i.e., add standard deviation $\sigma=4$mm clipped at $2\sigma$) to stress test recovery from failures via replanning with grounded closed-loop feedback. Results are presented in Table 2.

| | **LLM** | **+Inner Monologue** | | |
|---|---|---|---|---|
| **Task Family** | *Object* | *Object* | *Success* | *Object + Success* |
| Finish 3-block stacking | 20% | 40% | 40% | **100%** |
| Sort fruits from bottles | 20% | 50% | 40% | **80%** |
| **Total** | 20% | 45% | 40% | **90%** |

**Table 2:** Success rates averaged across 10 runs in real-world pick-and-place. We observe significant improvement in Inner Monologue with *Object* and *Success* feedback, with the two feedback being complementary to each other.

**Analysis.** We compare to variants with only *Object* or *Success* feedback, as well as an open-loop variant ("LLM Object") that only runs object recognition once at the beginning of the task (similar to the system demonstrated in [19]). The partial 3-block stacking task highlights an immediate failure mode of the open-loop baseline, where the initial scene description struggles to capture a complete representation of the scene (due to clutter and occlusion) to provide as input to the multi-step planner. As a result, the system only executes one pick-and-place action – and cannot recover from mistakes. To address these shortcomings, Inner Monologue (*Object + Success*) leverages closed-loop scene description and success detection after each step, which allows it to successfully replan and recover from policy mistakes.

## 4.3 Real-World Mobile Manipulator in a Kitchen Setting

We implement Inner Monologue in a robotic system using the kitchen environment and task definitions described in SayCan [21]. The Everyday Robots robot, a mobile manipulator with RGB observations, is placed in an office kitchen to interact with common objects using concurrent [99] continuous closed-loop control.

The baseline, SayCan [21], is a method that plans and acts in diverse real world scenarios by combining an LLM with value functions of control policies. While SayCan creates plans that are grounded by the affordances of value functions, the LLM predictions in isolation are never given any closed-loop feedback.

We use an instantiation of Inner Monologue that uses (i) PALM [8] for planning, (ii) value functions from pre-trained control policies for affordance grounding [21], (iii) a learned visual classification model for *Success* feedback, (iv) human-provided *Object* feedback, and (v) pre-trained control policies for relevant skills in the scene. We also perform a case study where we allow the agent to ask questions and source *Human* feedback directly; results are shown in Fig 5a and the Appendix.

We evaluate on 120 runs over three task families: (1) four manipulation tasks, (2) two dexterous manipulation tasks utilizing drawers, and (3) two long-horizon combined manipulation and navigation tasks. We consider both cases with and without manually-added adversarial disturbances during control policy executions that cause skill policy rollouts to fail. While these failures occur naturally even without perturbations, the adversarial disturbances creates a consistent comparison between methods that requires retrying or replanning to accomplish the original instruction.

| Task Family | SayCan | +Inner Monologue | |
| | | *Success* | *Object + Success* |
| --- | --- | --- | --- |
| **No Disturbances** | | | |
| Manipulation | 50.0% | 62.5% | **75.0%** |
| Mobile Manipulation | 50.0% | 50.0% | **75.0%** |
| Drawers | 83.3% | 83.3% | **100.0%** |
| **With Disturbances** | | | |
| Manipulation | 12.5% | 25.0% | **33.3%** |
| Mobile Manipulation | 0.0% | 25.0% | **75.0%** |
| Drawers | 0.0% | **44.4%** | **44.4%** |
| Total | 30.8% | 48.7% | **60.4%** |

**Table 3:** Averaged success rate across 120 evaluations on several task families in our real-world mobile manipulation environment. We consider a standard setting and adversarial setting with external human disturbances. In all cases, LLM-informed embodied feedback is shown to be effective in improving robustness of the system, especially when low-level policies are prone to failures.

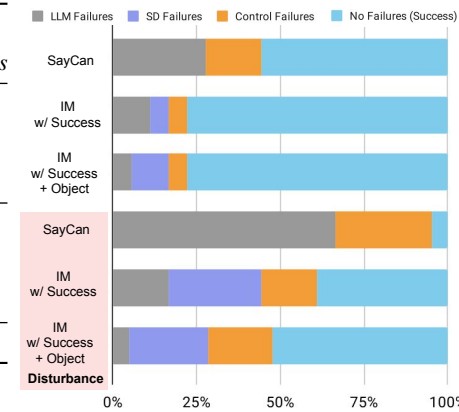

**Figure 4:** Failure causes on 120 evaluations. When disturbances are added (red), only the Inner Monologue variants consistently complete the instructions.

**Analysis.** Without adversarial disturbances, the baseline SayCan performs reasonably on all tasks, yet incorporating LLM-informed feedback in Inner Monologue allows further improvement by effectively retrying or replanning under natural failures. The most notable difference is in the cases with adversarial disturbances. Without any LLM-informed feedback SayCan has success rate close to 0% since LLM always assume successful execution of previous skills. Inner Monologue significantly outperforms SayCan because of its ability to invoke appropriate recovery modes depending on the environment feedback. Analysis on the failure causes indicates that *Success* and *Object* feedback can reduce LLM planning failures and thus overall failure rate, albeit at the cost of introducing new failure modes to the system.

## 4.4 Plan Generalization Capabilities

Although LLMs can generate fluent continuation from the prompted examples, we surprisingly find that, Inner Monologue demonstrates many impressive reasoning and replanning behaviors beyond the examples given in the prompt. Using a pre-trained LLM as the backbone, the method also inherits many of the appealing properties from its versatility and general-purpose language understanding. In this section, we demonstrate a few of these capabilities; additional capabilities are shown in Appendix (Fig **??** and Fig **??**).

**Continued Adaptation to New Instructions.** Although not explicitly prompted, the LLM planner can react to human interaction that changes the high-level goal mid-task. Fig 5a demonstrates a challenging case, where *Human* feedback changes the goal during the plan execution, and then changes the goal yet again by saying "finish the previous task". In another instance, despite not being explicitly prompted to terminate after a human says "please stop", the LLM planner generalizes to this scenario and predicts a "done" action.

**Self-Proposing Goals under Infeasibility.** Instead of mindlessly following human-given instructions, Inner Monologue can also propose alternative goals to achieve when the previous goal becomes infeasible. In Fig 5b, to solve the task "put any two blocks inside the purple bowl", while the first attempted block is intentionally made too heavy for the robot, Inner Monologue proposes to "find a lighter block" and successfully solves the task.

**Multilingual Interaction.** Pre-trained LLMs are known to be able to translate from one language to another, without any finetuning. We observe that such multilingual understanding also transfers to the embodied settings. Fig 5c shows a case when an instruction is in Chinese, the LLM planner can still correctly interpret it, re-narrate it as a concrete goal to execute in English, and accordingly replan its future actions. Occasionally, we find that this capability even extends to symbols and emojis.

**Retrospective Scene Understanding.** We also observe that Inner Monologue demonstrates retrospective scene understanding based on past actions and environment feedback, which requires temporal and embodied reasoning. In Fig 5d, after series of actions, we can turn to ask questions about the resulting scene, again a structure that has not appeared in the prompt.

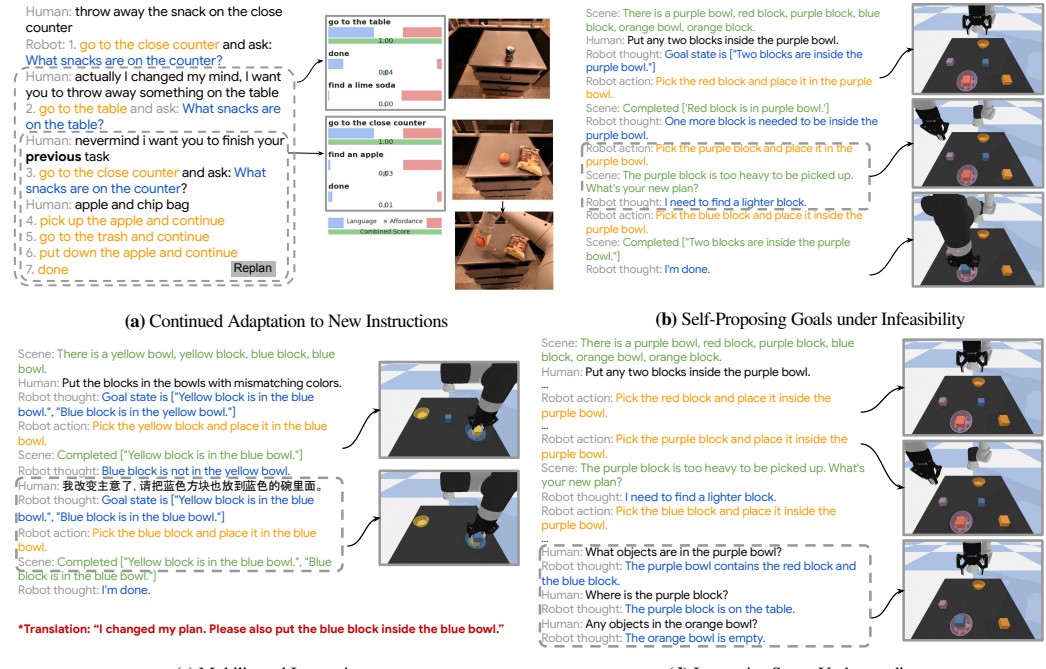

**(a)** Continued Adaptation to New Instructions

**(b)** Self-Proposing Goals under Infeasibility

**(c)** Multilingual Interaction

**(d)** Interactive Scene Understanding

**Figure 5:** Informing LLM with embodied feedback enables many generalization capabilities, all of which are achieved without similar prompted examples. For instance, Inner Monologue can continually adapt to new instructions given by humans, propose new goals to achieve when faced with infeasibility for the previous plan, interact with humans in different natural languages, and answer questions about the current scene given past actions and feedback.

Despite the appealing findings about these generalization capabilities, we observe that they are of varying levels of consistency when no similar examples have been provided in the prompt, likely limited by the current capabilities of the language models. However, we believe that further investigations into these behaviors and addressing their limitations would each lead to exciting future directions.

## 5 Conclusions, Limitations & Future Works

In this work, we investigated the role that environment feedback plays for LLMs reasoning in tasks involving embodied robotic planning and interaction. We presented a general formulation Inner Monologue that combines different sources of environment feedback with methods fusing LLM planning with robotic control policies and studied its instantiations in three distinct domains. We found that environment feedback significantly improves high-level instruction completion, especially in challenging scenarios with adversarial disturbances. Finally, we analyze generalization capabilities of Inner Monologue that highlight how closed-loop language feedback enables replanning even in complex unseen settings.

**Limitations.** In Sec 4.1 and Sec 4.3, we assume access to oracle scene descriptors in the form of human observers or scripted systems to provide textual description back to the LLM planner. We study the viability of learned systems scene description and object recognition in Appendix Table **??**. As for failure modes, Inner Monologue may fail due to several sources of errors: (1) success detections, (2) LLM planning errors, and (3) control errors. False negative predictions from the success detector lead to additional retry attempts, while false positive predictions add adversarial partial observability to the environment. In some instances, we found that the LLM planners ignored the environment feedback and still proposed policy skills involving objects not present in the scene. Additionally, the performance of low-level control policies limits not only overall high-level instruction completion performance, but also limits the scope of tasks that the LLM is able to plan actions for.

**Future Works.** Several fronts can be improved by future works. First, with advances in image/video captioning and visual-question answering, a fully automated system of Inner Monologue can be implemented without a human in the loop as an oracle. Second, improvements can be made on how to aggregate potentially inaccurate sources of information, such as using text to describe the uncertainty of the feedback modules, or including additional feedback modules for safety and ethics for the proposed plans. Finally, enabling low-level control policies to take as input the textual feedback by LLM also leads to exciting future directions.

**Acknowledgments**

The authors would like to thank Kanishka Rao and Vincent Vanhoucke for valuable feedback and discussions. In addition, the authors would like to acknowledge the large team who built [21], upon which we construct our Kitchen Mobile Manipulation experiments.

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
