# OpenReview forum: "Inner Monologue: Embodied Reasoning through Planning with Language Models"
_robot-learning.org/CoRL/2022/Conference — CoRL 2022 Poster_

### Official Review · Reviewer_YS48 · 2022-08-01

**Originality:** Excellent
**Technical Quality:** Excellent
**Clarity Of Presentation:** Excellent
**Impact:** 4

**Recommendation:**

Weak Accept: I recommend accepting the paper, but will not argue for my recommendation if the majority of other reviewers have a different opinion.

**Summary:**

This work extends prior algorithms for robotic planning with LLMs to incorporate feedback, which increases tasks success rate in presence of imperfect low-level control and a dynamic environment.

**Issues:**

 - A key limitation of the approach in my opinion is in action failure handling, which only retries actions. However, feedback from the environment could also require major replanning. For instance, failure to throw an object into the trash because it was full, or failure to open a drawer because it is locked. In my opinion, studying "methods and sources of feedback ... and their impact on downstream execution success" should go in tandem with studying methods for handling such feedback. While the rest of the system demonstrated different implementations by swapping out the LLM, hardware, low-level action policies, the prompts in the appendix implies that Inner Monologue only retries actions until it is successful.

**Quality Of The Limitations Section:**

Limitations are addressed clearly

**Reviewer Expertise:**

5: The reviewer is absolutely certain that the evaluation is correct and very familiar with the relevant literature

**Robotics Focus:**

Sufficient demonstration on hardware

**Strengths And Weaknesses:**

Strength:
- The approach is simple and effective.
- The types of feedback considered are diverse.
- The various implementations of the idea using different LLMs, feedback mechanisms, primitive descriptions/language encoding, low-level control primitives in different domains demonstrates the generality of the idea.
- Video does a great job at motivating and describing the approach, as well as demonstrating the system's capabilities.
- The figures are informative.

Weakness:
-  The key weakness in their goal of studying how environment feedback helps planning is in their lack of investigation into feedback handling mechanisms (such as high level replanning). Their experiments only demonstrate retrying until actions are successful. More details in issues section.
- The key weakness in making the system a fully autonomous language-conditioned system is its reliance on oracle (human or scripted systems) feedback. However, this weakness does not affect their goal in studying how environment feedback helps planning.



**Summary Of Recommendation:**

Inner Monologue demonstrates a diverse set of systems implementing their key idea of environment feedback to LLM planner. All components of their systems are comprehensively studied except for feedback handling behaviour, which in my opinion is a crucial aspect in studying environment feedback and its impact on downstream action success.

---

> ### Author Response · Authors · 2022-08-28
> **Response for Reviewer YS48**
>
> **Comment:**
>
> We thank the reviewer for recognizing that the method being “simple and effective” and that we benchmark diverse sets of feedback in multiple domains. We address the issues raised by the reviewer below:
>
> >*"The key weakness in their goal of studying how environment feedback helps planning is in their lack of investigation into feedback handling mechanisms (such as high level replanning).  Their experiments only demonstrate retrying until actions are successful... A key limitation of the approach in my opinion is in action failure handling, which only retries actions. However, feedback from the environment could also require major replanning."*
> - We would like to first point out that the system does not only retry actions but also replans (completely changing its future actions) based on the provided closed-loop feedback. For example, as shown in the [walkthrough video (06:15)](https://inner-monologue.github.io/img/im_supp_video_corl_compressed.mp4) on the accompanying website included in the submission, the planner is able to reason about complex human feedback and change its plan entirely based on human preferences and object list, all provided in a closed-loop fashion. The similar replanning behaviors are also studied and observed in other domains that we investigate in this work. We apologize for causing this misunderstanding and will update the draft accordingly.
>
> - To further showcase the replanning capabilities of the system, we are also adding the following flowchart to the main manuscript to show how closed-loop feedback influences the decisions of the LLM planner. Please see the flowchart [here](https://inner-monologue.github.io/img/flowchart_decision_making.png), which we also attach as part of the response for convenience.
>
> >*"The key weakness in making the system a fully autonomous language-conditioned system is its reliance on oracle (human or scripted systems) feedback. However, this weakness does not affect their goal in studying how environment feedback helps planning."*
>
> - We want to clarify that the human feedback used in our work is just a part of study, a significant part of the system can run autonomously, in fact, all the results we show in our benchmarks are running autonomously (Table 1, 2, and 3). Second, we are not sure what scripted systems refer to, our success detection, scene description and object detection are mostly learned rather than scripted.
>
> - We thank the reviewer for recognizing the main goal of our work is to study how LLM planning can incorporate different types of environment feedback, as opposed to studying how to implement the feedback mechanisms themselves. Nonetheless, we do want to clarify that while it is true that while some of our feedback sources were good enough to be considered oracles, that the majority of feedback systems in our experiments were autonomous. For instance, we utilize an autonomous Object feedback system based on MDETR in the Real Tabletop Environment (Section 4.2) and a data-driven Success feedback signal based on CLIP in the Real Kitchen Environment (Section 4.3).
>
> - We only assume 2 oracle sources of feedback: Object and Human feedback in the Real Kitchen Environment. For the Object feedback system, we discuss in Appendix A.3 how the oracle feedback could be swapped with learned object recognition systems like MDETR or VILD and show their performance in Table 5. We utilize an oracle Object implementation instead of these learned models in order to better study our main focus of this work, but plan on improving these autonomous implementations for the final manuscript. The other oracle feedback we assume is Human feedback, but it is not strictly required. In general, our system is not specific to a set of feedback but can work with any subset of feedback; furthermore, it does not require retraining to extend to any types of new feedback as long as they are expressed in language.
>
>
> **Zip File:**
>
> /attachment/5f0d4bbd7b1baf90710bd9a0371c4e705e69c78c.zip

---

### Official Review · Reviewer_JWy9 · 2022-08-01

**Originality:** Very Good
**Technical Quality:** Very Good
**Clarity Of Presentation:** Very Good
**Impact:** 4

**Recommendation:**

Weak Accept: I recommend accepting the paper, but will not argue for my recommendation if the majority of other reviewers have a different opinion.

**Summary:**

This paper proposes using language as a form of embodied feedback for LLM-based robot planning. Three types of environment feedback, i.e. success detection, passive and active scene descriptions, are expressed in language to show how they improve the success rate for planning in simulated and real-world environments.

**Issues:**

It would be helpful if the authors can address the following issues.
- Discuss how to extend the language feedback beyond LLM-based planners.
- Clarify how to determine which feedback to be active at each time step.
- Minor:
    - On line 218, this paper mentioned an open-loop variant, which one is it in table 2?
    - What is an LLM error in fig 6? Does it mean that the LLM planner generates the wrong steps? If so, I’m surprised that SayCan has such significant increase of LLM failures when there is disturbances.
    - The line hight in this paper is considerably smaller than other papers.

**Quality Of The Limitations Section:**

Limitations are addressed clearly

**Reviewer Expertise:**

5: The reviewer is absolutely certain that the evaluation is correct and very familiar with the relevant literature

**Robotics Focus:**

Sufficient demonstration on hardware

**Strengths And Weaknesses:**

Strengths
- This is the first study that incorporate environment feedback into LLM-based planner. This indicates it is possible to use LLM to represent the knowledge of the robot and reason over the it with feedback represented in language which opens ways to incorporate LLMs as reasoning models for task planning.
- This paper evaluates different types of feedback in both simulated and real world domains to show the feasibility of the proposed ideas to how it can improve reasoning in long-horizon tasks.

Weakness
- It is unclear which type of feedback should be active at each time step. It seems that the selection of feedback is hand-crafted for different types of planning steps. But sometimes the hand-crafted feedback may not be sufficient. For example, if the robot fails to put the blue block in the blue tray, the success detector says false, but the block is dropped to a red try, therefore, another scene detector needs to recognize this and inform the planner that the blue block is in the red tray. So, when to instantiate a type of feedback and how to combine multiple feedback are important to utilize feedback.
- Each experiment only includes a subset of feedback. For example, scene feedback is not used in real world experiments. It is unclear why those types of feedback are omitted.
- The planner is LLM-based. So it is natural to incorporate feedback as part of the language prompt. It will be very interesting if this language feedback can be incorporated into other types of planners too.

**Summary Of Recommendation:**

This paper presents an interesting case study to incorporating environment feedback into LLM-based planners and shows that these feedback can help task success rate. However, the current implementation still requires heuristics to design which feedback to apply for each planning step which may not utilize the available feedbacks.

---

> ### Author Response · Authors · 2022-08-28
> **Response for Reviewer JWy9 (Part 1)**
>
> We thank the reviewer for recognizing our work as the first study that incorporates environment feedback into LLM-based planner, and agree on the potential of the proposed method to enable embodied reasoning in long horizon tasks.
>
> >*"It is unclear which type of feedback should be active at each time step. It seems that the selection of feedback is hand-crafted for different types of planning steps."*
>
> - We would like to point out that the process of when we trigger each feedback at each timestep is not hand-crafted. However, we thank the reviewer for pointing out the lack of clarity and we will update the paper accordingly. Regarding when each feedback is triggered, for human feedback, it is up to LLM to decide when to ask a clarifying question if a command is ambiguous. For other types of feedback, we define a “canonical” order of when to trigger each type of feedback for each domain. For example, in the simulated tabletop domain, the “Object” feedback is provided at the beginning of the task to indicate the detected objects. However, we would also like to point out that LLM is observed to be robust to new ordering of feedback unseen from the given prompt, as we studied this in Section 4.4 of the paper.
>
> >*"For example, if the robot fails to put the blue block in the blue tray, the success detector says false, but the block is dropped to a red tray, therefore, another scene detector needs to recognize this and inform the planner that the blue block is in the red tray."*
>
> - For this particular failure case the reviewer brought up, we believe it is addressed by "task-progress scene description" proposed in the paper. Here is the exact rollout of this failure case in the tabletop simulation domain.
> ```
> Scene: There is a red block, red bowl, blue block, blue bowl.
> Human: Put the block in the bowl with matching color.
> Robot thought: Goal state is ["Red block is on the red bowl.", "Blue block is on the blue bowl."]
>
> Robot action: Pick the blue block and place it on the blue bowl.
> Scene: You have completed ["Blue block is on the red bowl."]
> Robot thought: Blue block is in the wrong bowl. It should be in the blue bowl.
>
> Robot action: Pick the blue block and place it on the blue bowl.
> Scene: You have completed ["Blue block is on the blue bowl."]
> Robot thought: Now I only need to put the red block in the red bowl.
>
> Robot action: Pick the red block and place it on the red bowl.
> Scene: You have completed ["Red block is on the red bowl.", "Blue block is on the blue bowl."]
> Robot thought: I'm done.
> ```
>
> >*"So, when to instantiate a type of feedback and how to combine multiple feedback are important to utilize feedback… Each experiment only includes a subset of feedback. For example, scene feedback is not used in real world experiments. It is unclear why those types of feedback are omitted."*
>
> - Our general observation is that combining all sources of feedback leads to the best performance. And the planner is robust to occasionally missing a subset of feedback or changes to feedback order. However, due to the practical consideration that certain types of feedback involve a manual process, such as “Scene” and “Human” feedbacks, we do not study them across all domains.
> - Qualitatively, we find that “Object” feedback is most important while other different types of feedback are useful in specific scenarios. For example, neither “Object” feedback nor “Success” feedback alone are sufficient in the real-world tabletop domain; only the combination of both forms of feedback - a feedback we refer to as “task-progress scene description” - enable solving the task. Regarding how different feedbacks provide separate but critical roles in informing the LLM planner, we provide a flowchart showing how each feedback influences the planner in the response to the meta-reviewer.
> - Having said this, we would like to reiterate that our main contribution in this work is not to determine the optimal or minimal feedback under a given robotic setting, but rather to provide a study on how LLMs are generally able to incorporate different types of feedback, as long as they can be sufficiently expressed in text. Furthermore, the system does not require retraining to extend to new types of feedback.

---

> ### Author Response · Authors · 2022-08-28
> **Response for Reviewer JWy9 (Part 2)**
>
> >*"The planner is LLM-based. So it is natural to incorporate feedback as part of the language prompt. It will be very interesting if this language feedback can be incorporated into other types of planners too. Discuss how to extend the language feedback beyond LLM-based planners."*
>
> - We agree that this is an interesting and important direction, though we consider it for now out of scope of our current work. That being said, we are particularly intrigued by the recent performance of LLMs for coding that indicate how an LLM can integrate and interact with more structured systems outside of natural language. For planning, we expect this could take the form of structuring problems for optimizers, providing guidance to planners, or modeling for instance. We think this is an exciting direction and that Inner Monologue shows foundations of how LLMs can reason with feedback.
>
> >*"On line 218, this paper mentioned an open-loop variant, which one is it in table 2?"*
>
> - The open-loop variant is the column titled "LLM Object". It is called open-loop because it doesn’t have success detection feedback. We will update the paper draft to avoid any confusion.
>
> >*"What is an LLM error in fig 6? Does it mean that the LLM planner generates the wrong steps? If so, I’m surprised that SayCan has such significant increase of LLM failures when there is disturbances."*
>
> - We apologize for the confusion: we refer to incorrect plans generated by LLM as “LLM errors”. An example would be the following. Given the task “bring me a bottle of water”, the LLM planner will plan and execute 1. Find a bottle of water, 2. Pick up the bottle of water, 3. Bring it to you, 4. Done. Imagine at the second step, the manipulation action failed under disturbance. In this case, the correct LLM plan is to try “Pick up the bottle of water again”.  But the SayCan method won’t be able to know there is a failure. So it will continue to plan the next step “Bring it to you”, hence resulting in an “LLM error”. When adding feedback, those failures are significantly reduced, as seen in Figure 6. We will update the draft to avoid using the term to resolve any confusion.
>
> >*"The line height in this paper is considerably smaller than other papers."*
>
> - Thanks for the note, we will make sure to fix it in the camera ready version.

---

### Official Review · Reviewer_Ywcr · 2022-08-11

**Originality:** Fair
**Technical Quality:** Fair
**Clarity Of Presentation:** Very Good
**Impact:** 3

**Recommendation:**

Weak Reject: I recommend rejecting the paper, but will not argue for my recommendation if the majority of other reviewers have a different opinion.

**Summary:**

The paper proposes a system for using natural language feedback generated by large language models(LLMs) to improve task planning in embodied environments. The natural language feedback is generated by using auxiliary models for success detection and scene description and prompting the LLM with information from their outputs. The paper demonstrates the effectiveness of the system on a good set of real-world and simulated environments of a robot performing different tasks.

**Issues:**

Issues are listed in the weaknesses.

**Quality Of The Limitations Section:**

Additional details required

**Reviewer Expertise:**

5: The reviewer is absolutely certain that the evaluation is correct and very familiar with the relevant literature

**Robotics Focus:**

Sufficient demonstration on hardware

**Strengths And Weaknesses:**


**Strengths:**
- The system does not require training a specialized model for the task and uses existing pre-trained models prompted with information from the environment.
- The benefit of updating the plans using specialized feedback from the auxiliary models is clear.

**Weakness**
- The idea of using multiple large pre-trained models (of vision and language) in a zero-shot manner via multimodal-informed prompting for successful task completion has been studied by Zeng et al. 2022. However, Zeng et al. do not demonstrate its use for robotics tasks.  What aspect of the system required a change when applied to a new domain?

- The importance of feedback for long horizon tasks is emphasized throughout the paper. However, no baseline verifies this need for increased feedback for longer tasks. A plot comparing models with and without feedback versus the length of the trajectory is useful. How many times in the course of a trajectory was the model able to benefit from feedback?

- Missing Prior Work:
   - There is no mention of prior work on self-talk [1] or of papers that follow up on the same idea.
   - There is no mention of work on the use of natural language feedback in robotic systems.

- No comparison against better decoding methods. An important baseline to look at in the case of fixing robot task plans is to 1) decode multiple possible top-scoring plans and 2) execute the next highest scoring plan in the case of a failure. However, there is no comparison with such a baseline.

- Failure modes of the large language model planner: The limitation section describes how the model ignores feedback from the environment several times. What do some of these cases look like? What are recurrent patterns of failures?

- "Emergent" human feedback processing:
There is nothing emergent about the model's ability to use human feedback. The feedback is still in the form of natural language, very much in the domain of prompting the LLM. The property of the model to be able to process human feedback to modify its suggested plan is hence not an emergent property but one to expect given the experiments on using specialized cues from the environment.

[1] Unsupervised Commonsense Question Answering with Self-Talk(https://aclanthology.org/2020.emnlp-main.373) (Shwartz et al., EMNLP 2020)

**Summary Of Recommendation:**

While using natural language feedback generated by an LLM prompted with specialized environment information is interesting. The technical novelty of the paper is limited.
1. The idea of using multiple large pre-trained models (of vision and language) in a zero-shot manner via multimodal-informed prompting for successful task completion has been studied by Zeng et al. 2022.
2. The idea of using "inner-monologue" is not new. Prior work in NLP has proposed the use of "self-talk" as a system for corrections.
3. The paper is missing important baselines and analysis.

---

> ### Author Response · Authors · 2022-08-18
> **Quick clarification question about the decoding baseline**
>
> We thank the reviewer for recognizing one of the core themes of our paper, that “the benefit of updating the plans with specialized feedback from auxiliary models is clear” especially when using “existing pre-trained models” that do not require additional training. In addition, thank you for posing interesting questions and suggestions that will help make our work stronger!
>
> First, we would like to point a few important distinctions with Socratic Models (Zeng et al 2022). Socratic Models uses feedback loops between Language Models, the total number of steps is fixed, and depends on the task. We lift this restriction in Inner Monologue, where the number of steps/feedback loops is variable and subject to change depending on observed outcomes. This is driven by the LM planner, whose generated plans serve as a kind of "task success progress bar". This is a key distinction because in the tasks benchmarked in SMs (e.g. image captioning, video-to-text retrieval), LMs are used as "passive observers". All information needed for the task is presented upfront, and it is up to the LMs to digest it all to propose a task-specific output,  whereas in Inner Monologue, specifically because we are interested in "embodied" systems, our modules serve as "active explorers". New information can come in at any point in time (something succeeded or failed, or new info about the scene is revealed). This is a dynamic new aspect that makes our problem domain (embodiment via robotics) interesting to investigate.
>
> Second, we plan on carefully updating our paper based on your review, but first wanted to clarify your comments about decoding baselines. In our existing LM planning setup, we used a closed loop setting where the LM plans only the immediate next skill to choose. Upon a failure, the LM is allowed to choose how to “fix” the plan, which may include retrying the skill that failed; in contrast, an open loop setting would at the start of the episode score multiple complete sequences of skills to execute. While we do consider the open loop setting as a baseline (see experiments in 4.2 and 4.3), it is naturally weaker than the closed loop variants. To study the baseline you suggest, we hope to implement the following baseline: “In a closed-loop setting, score multiple skills each step. Upon a failure, execute the next highest scoring step.”
>
> Does this accurately represent the baseline you proposed?

---

> ### Author Response · Authors · 2022-08-28
> **Response for Reviewer Ywcr (Part 1)**
>
> Thank you for your patience! We have finalized the additional experiments you requested and can provide some additional analysis based on your feedback.
>
> >*"A plot comparing models with and without feedback versus the length of the trajectory is useful"*
>
> - We appreciate the suggestion that we should clarify the importance of feedback as task horizons increase. We’ve provided the requested plot that analyzes the importance of feedback as trajectory length increases in the Real Kitchen Environment across 123 evaluations. We find that feedback is important when tasks become more challenging and there is an increased risk of failures that require retrying or replanning. These occur when the trajectory length increases or when adversarial disturbances are applied. We note that the baseline especially suffers in the latter case, and is unable to solve any long tasks under disturbances. We will update the final version of the paper with these details.
>
> |                  | Disturbance | SayCan | SayCan + Feedback |
> |------------------|:-----------:|:------:|:-----------------:|
> | All 2 step tasks |      No     | 54.55% |       75.00%      |
> | All 4 step tasks |      No     | 44.44% |       70.59%      |
> | All 2 step tasks |     Yes     | 18.18% |       45.83%      |
> | All 4 step tasks |     Yes     |  0.00% |       41.18%      |
>
> >*"The idea of using multiple large pre-trained models (of vision and language) in a zero-shot manner via multimodal-informed prompting for successful task completion has been studied by Zeng et al. 2022."*
>
> - We provide discussions on the Socratic Models by Zeng et al. 2022 in our initial response above. To summarize, a key difference is that Socratic Models does not study the challenges particular to robots operating in dynamic environments where closed-loop environment feedback is essential to the planner. Although it is certainly expected that closed-loop environment feedback can improve the planner, to the best of our knowledge, it hasn’t been studied how such system can be built or quantified how much value the feedback brings to the planner.
>
> - In addition to our initial response, we would like to further elaborate on how such extension to the robotics setting is non-trivial. One main challenge when applying Socratic Models to robotics directly is that pre-trained models are often not grounded to relevant robotics settings where skills or tasks have very specific and domain-specific meanings. For example, to the best our knowledge, existing pre-trained VQA models are incapable of performing success detections, which require attention to subtle visual details relevant to an action. As a result, we need to finetune CLIP on domain-specific data to provide meaningful success detection results. Furthermore, LLM is often incapable of consistently grounding action plans in an open-world scenario, such as our real-world kitchen environment. Therefore, we utilize the affordance-grounding method introduced in SayCan to ensure that plans proposed by general pre-trained models are achievable by the skills the robot has.

---

> ### Author Response · Authors · 2022-08-28
> **Response for Reviewer Ywcr (Part 2)**
>
> >*"Missing Prior Work. There is no mention of prior work on self-talk [1] or of papers that follow up on the same idea. There is no mention of work on the use of natural language feedback in robotic systems."*
>
> - Thank you for pointing this out, we’ve added citations and discussions to Self-Talk as well as many other works (listed below) that we found relevant. If there are any others the reviewer finds particularly relevant we’d be happy to add discussions to them.
>
> [1] M. Dorigo and M. Colombetti. Robot shaping: Developing autonomous agents through learning.
> Artificial intelligence, 71(2):321–370, 1994.
>
> [2] K. Judah, S. Roy, A. Fern, and T. G. Dietterich. Reinforcement learning via practice and critique
> advice. In Twenty-fourth AAAI conference on artificial intelligence, 2010.
>
> [3] R. Loftin, B. Peng, J. MacGlashan, M. L. Littman, M. E. Taylor, J. Huang, and D. L. Roberts.
> Learning behaviors via human-delivered discrete feedback: modeling implicit feedback strate-
> gies to speed up learning. Autonomous agents and multi-agent systems, 30(1):30–59, 2016.
>
> [4] A. L. Thomaz, G. Hoffman, and C. Breazeal. Reinforcement learning with human teachers:
> Understanding how people want to teach robots. In ROMAN 2006-The 15th IEEE International
> Symposium on Robot and Human Interactive Communication, pages 352–357. IEEE, 2006.
>
> [5] A. Najar, O. Sigaud, and M. Chetouani. Interactively shaping robot behaviour with unlabeled
> human instructions. Autonomous Agents and Multi-Agent Systems, 34(2):1–35, 2020.
>
> [6] G. Kuhlmann, P. Stone, R. Mooney, and J. Shavlik. Guiding a reinforcement learner with natural
> language advice: Initial results in robocup soccer. In The AAAI-2004 workshop on supervisory
> control of learning and adaptive systems. San Jose, CA, 2004.
>
> [7] M. Ahn, A. Brohan, N. Brown, Y. Chebotar, O. Cortes, B. David, C. Finn, K. Gopalakrish-
> nan, K. Hausman, A. Herzog, et al. Do as i can, not as i say: Grounding language in robotic
> affordances. arXiv preprint arXiv:2204.01691, 2022.
>
> [8] J. D. Co-Reyes, A. Gupta, S. Sanjeev, N. Altieri, J. Andreas, J. DeNero, P. Abbeel, and
> S. Levine. Guiding policies with language via meta-learning. ICLR, 2018.
>
> [9] P. Sharma, B. Sundaralingam, V. Blukis, C. Paxton, T. Hermans, A. Torralba, J. An-
> dreas, and D. Fox. Correcting robot plans with natural language feedback. arXiv preprint
> arXiv:2204.05186, 2022.
>
> [10] V. Shwartz, P. West, R. L. Bras, C. Bhagavatula, and Y. Choi. Unsupervised commonsense
> question answering with self-talk. arXiv preprint arXiv:2004.05483, 2020.
>
> [11] L. Reynolds and K. McDonell. Prompt programming for large language models: Beyond the
> few-shot paradigm. In Extended Abstracts of the 2021 CHI Conference on Human Factors in
> Computing Systems, pages 1–7, 2021.
>
> [12] J. Wei, X. Wang, D. Schuurmans, M. Bosma, E. Chi, Q. Le, and D. Zhou. Chain of thought
> prompting elicits reasoning in large language models. arXiv preprint arXiv:2201.11903, 2022.
>
> [13] T. Kojima, S. S. Gu, M. Reid, Y. Matsuo, and Y. Iwasawa. Large language models are zero-shot
> reasoners. arXiv preprint arXiv:2205.11916, 2022.

---

> ### Author Response · Authors · 2022-08-28
> **Response for Reviewer Ywcr (Part 3)**
>
>
> >*"No comparison against better decoding methods."*
>
> - We implement our proposed interpretation of a competitive decoding baseline from the initial response and provide the results below in both the **Simulated Tabletop Domain** and the **Real-World Mobile Manipulation Domain**.
>
> - **Simulated Tabletop**: We implement the baseline as follows: each time when we sample the next action, we draw 10 samples from LLM and execute the most likely action as calculated by mean log probability under LLM. In case of a failure, the next most likely action is executed until it is successful as indicated by the success detector.
>
> - As shown below, while the proposed decoding strategy is competitive, it is generally outperformed by Inner Monologue, and the most significant gap is on tasks that involve combinatorial state space, e.g. “stack all the blocks”. To complete these tasks in case of an action failure, replanning is often needed while retrying is not enough. However, drawing multiple samples from LLM without providing textual feedback often does not lead to replanning behaviors because the majority of the samples are the same or very similar to each other. Therefore, executing the next most likely action often amounts to retrying the same action repeatedly until there is a success.
>
> |                                               | Next Best Decoding | IM (Object + Scene) |
> |-----------------------------------------------|:------------------:|:-------------------:|
> |                “Pick and place”               |       95.00%       |        94.00%       |
> |             “Stack all the blocks”            |        2.00%       |        26.00%       |
> |  “Put all the blocks on the [x] corner/side”  |       34.00%       |        30.00%       |
> |      “Put all the blocks in the [x] bowl”     |       42.00%       |        56.00%       |
> |   “Put all the blocks in different corners”   |       30.00%       |        26.00%       |
> |    “Put the blocks in their matching bowls”   |       78.00%       |        82.00%       |
> |      “Put the blocks on mismatched bowls”     |       70.00%       |        86.00%       |
> | “Stack all the blocks on the [x] corner/side” |        0.00%       |        6.00%        |
>
>
> - **Real-World Mobile Manipulation**: In this domain, we build upon the SayCan baseline: when a failure is detected by the success detector, we automatically have the planner select the second-best skill (ie. where the best skill is the step that just failed). However, unlike in the simulated tabletop domain, the second-best skill is strictly distinct from the first-best skill because skills are selected by scoring instead of generated autoregressively in SayCan.
>
> - As shown below, we find that in some scenarios, this Next Best Decoding baseline is able to recover when disturbances force replans that happen to align with the second best skill prediction. For example, for the instruction “can you throw away the soda on the table”, the policy went to find a coke can but we removed the coke can from the scene; the second best option was to pick up a redbull can and throw it away, which indeed satisfied the original instruction of throwing away a soda. However, we note that in general, the second best skill option does not reliably make progress towards the overall goal, and find that using closed-loop replanning with feedback from the environment is crucial to recovering from mistakes.
>
>
> |             | SayCan | Next Best Decoding | IM (Success) | IM (Success + Scene) |
> |-------------|:------:|:------------------:|:----------------:|:------------------------:|
> |   Standard  | 61.11% |       62.50%       |      66.67%      |          83.33%          |
> | Adversarial |  4.76% |       18.75%       |      33.33%      |          44.00%          |

---

> ### Author Response · Authors · 2022-08-28
> **Response for Reviewer Ywcr (Part 4)**
>
> **Comment:**
>
> >*"Failure modes of the large language model planner"*
>
> - Thanks for the question, here we list a few potential failure modes. In general the system is robust and the failures only happen occasionally.
>
> - **Failure mode 1**: Occasionally the LLM doesn’t fully acknowledge the sources of the feedback. Here the scene description mentioned pick up coke is not successful and grapefruit soda is in the scene but LLM still tries to pick it up. Please see the example [image](https://inner-monologue.github.io/img/failure_case1.png), also attached below for convenience.
>
> - **Failure mode 2**: The LLM might retry infinite times and we have to manually terminate the process. Note this particular failure case is with success detection only, and if we add scene description LLM will choose to pick the other type of chips after failing at kettle chips a few times, instead of going into infinite loop. Please see the example [image](https://inner-monologue.github.io/img/failure_case2.png), also attached below for convenience.
>
> - **Failure mode 3**: Failure might also occur for the feedback modules we built for specific domain. For example, in the real-world mobile manipulation domain, the success detector finetuned from CLIP is unable to understand the subtle visual details in certain scenarios. Specifically in the attached image, it believes that the "pick up sponge" skill was successful while it was not. Please see the example [image](https://inner-monologue.github.io/img/failure_case3.png), also attached below for convenience.
>
> >*"There is nothing emergent about the model's ability to use human feedback"*
>
> - In our case, our motivation for the “emergent capabilities” section is to study robot behaviors that we did not explicitly optimize for via our provided prompts. We agree that many of these concepts were likely covered in the original pre-trained LM training distributions (such as multilingual datasets), so calling these behaviors “emergent” may be overly broad. We would propose referring to this section instead as “Plan Generalization Examples” and clarifying that these generalization capabilities, while novel and useful for the robotics behaviors they enable, are still driven by the power of the LM. The different examples will still showcase how despite our engineered prompts not containing these capabilities, the LM is still able to interact with grounded environment feedback to provide useful robotic plans.
>
>
> **Zip File:**
>
> /attachment/545c9648ffd3686c255c1636656b70c9abf4be9d.zip

---

### Meta-Review · Area_Chair_zCeo · 2022-08-11

**Recommendation:** Accept (Poster)
**Confidence:** 5

**Metareview:**

This paper presents a closed loop planning system that leverages LLMs for providing candidate plans given the success of action execution and the world state. Central to the approach is querying the LLM given the instruction and text prompts capturing success of skill execution and a description of the scene.

This is a systems oriented paper demonstrating and evaluating the utility of a LLM for interleaved planning and execution. The paper is well organised and clearly written. The results highlight the utility of LLMs to enable task execution before an explicit learning of a task model (given primitive robot skills amenable to composition). Further, experiments demonstrate contextual plan adaptation and success goal reaching for simulated and real data sets.

The main limitation of the work was observed to be the specialised nature of feedback prompts necessary for successful task execution. A clarification was sought if the proposed system has a principled way to determining the necessary feedback and to what extent the feedback is to be adjusted for long-running plans. In response, the authors clarified that the feedback is not hand crafted. Further,  the authors articulated their contribution towards demonstrating  the ability of LLM-based planners to integrated multi-modal feedback without focusing on the specific problem of determine how much or how best such a feedback may be generated. In response to a reviewer’s suggestion, the authors provided a comparison of models with and without feedback with an increase in the task horizon.

Additionally, the reviewers inquired the degree to which plan recovery is possible in the proposed framework. In response, the authors drew attention to results where plan adaptation occurs when an object falls is no longer in the reachable workspace. The reviewers further requested for a comparison with a baseline that searches over a set of high-candidate plans. As suggested, the authors provided a comparison with a Next Best Decoding that uses the next most likely skill.

Overall, the inclusion of new experimental insights strengthens the paper. A clear articulation of the technical advance in relation to Zang et al. 2022 is also recommended in the main manuscript.

---

> ### Author Response · Authors · 2022-08-28
> **Response for Area Chair zCeo (Part 1)**
>
> **Comment:**
>
> We thank the reviewer for finding our paper to be “well organised and clearly written” and that our method “demonstrates contextual plan adaptation and success goal reaching for simulated and real data sets”.
>
> >*"Clarification is needed if the proposed system has a principled way to determining the necessary feedback and to what extent the feedback is to be adjusted for long-running plans."*
>
> - Our main contribution in this work is not to determine the optimal or minimal feedback under a given robotic setting, but rather to provide a study on how LLMs are generally able to incorporate different types of feedback. Qualitatively, we find that different types of feedback are useful in specific scenarios. For example, neither Object feedback nor Success feedback alone are sufficient in the Real-World Tabletop Rearrangement Task; only the combination of both forms of feedback enable solving the task. We provide a flowchart that demonstrates how different forms of feedback provide separate but critical roles in informing the LLM planner about different parts of the system. Please see the attached flowchart [here](https://inner-monologue.github.io/img/flowchart_decision_making.png), which we also attach as part of the response for convenience. We will also add this to the final version of the manuscript.
>
> - In addition to the qualitative differences between feedback sources, we do see that Object feedback is quite important across all domains and success feedback alone is not enough to solve tasks. As for how the role of feedback changes as task horizons increase, we provide an analysis in our response to Reviewer Ywcr.
>
> >*"asked if the system can deal with plan recovery that requires major change in the plan"*
>
> - We study how feedback can inform major changes in the plan by introducing adversarial disturbances in our Real-World Kitchen Mobile Manipulation experiments in Section 4.3. While some disturbances cause skill failures that can be fixed by re-attempting the previous plan, some disturbances cause the scene to semantically change such that an entire new plan needs to be proposed. For instance, consider the case where a robot is tasked with retrieving a soda. Robot failures that irrecoverably change the environment (dropping the coke on the floor such that it’s impossible to pick it up) would require changing the plan altogether to finding and retrieving a different soda. On the other hand, human “failures” such as human feedback that switches the human’s preference to a healthy drink would similarly require significant replanning from the robot to find and retrieve a different drink. Some examples of what these re-plans look like are shown in Listing 4 in the appendix.
>
> >*"if the approach can be compared with a baseline that searches over a set of high-candidate plans"*
>
> - We clarify the nature of open-loop vs. closed-loop planning in our initial response to Reviewer Ywcr. Specifically, our SayCan baseline in Section 4.3 already do search over a set of skills in closed-loop fashion at each step; searching over a set of open-loop plans as Reviewer Ywcr proposed would be a weak baseline that wouldn’t be able to dynamically adjust to changing conditions on the fly. Thus, we propose a stronger version of this baseline that still varies the decoding logic of how the LLM chooses the next most likely skill in a closed-loop fashion. We implement this baseline in the Simulated Tabletop Environment and the Real Kitchen Environment. Overall, the "Next Best Decoding" baseline is able to outperform the SayCan baseline but is still inferior to all Inner Monologue variants, which allow the LLM to propose and pick the immediate next skill upon being given LLM-informed feedback. Please find the results below:
>
> **Zip File:**
>
> /attachment/28d62d700855805acb2da2ae6503346f9a545e30.zip

---

> ### Author Response · Authors · 2022-08-28
> **Response for Area Chair zCeo (Part 2)**
>
>
> **Simulated Tabletop**:
>
> |                                               | Next Best Decoding | IM (Object + Scene) |
> |-----------------------------------------------|:------------------:|:-------------------:|
> |                “Pick and place”               |       95.00%       |        94.00%       |
> |             “Stack all the blocks”            |        2.00%       |        26.00%       |
> |  “Put all the blocks on the [x] corner/side”  |       34.00%       |        30.00%       |
> |      “Put all the blocks in the [x] bowl”     |       42.00%       |        56.00%       |
> |   “Put all the blocks in different corners”   |       30.00%       |        26.00%       |
> |    “Put the blocks in their matching bowls”   |       78.00%       |        82.00%       |
> |      “Put the blocks on mismatched bowls”     |       70.00%       |        86.00%       |
> | “Stack all the blocks on the [x] corner/side” |        0.00%       |        6.00%        |
>
>
> **Real Kitchen**:
>
> |             | SayCan | Next Best Decoding | IM (Success) | IM (Success + Scene) |
> |-------------|:------:|:------------------:|:----------------:|:------------------------:|
> |   Standard  | 61.11% |       62.50%       |      66.67%      |          83.33%          |
> | Adversarial |  4.76% |       18.75%       |      33.33%      |          44.00%          |